# Social Factors Contributing to Healthcare Service Requirements during the First COVID-19 Lockdown among Older Adults

**DOI:** 10.3390/healthcare10101854

**Published:** 2022-09-23

**Authors:** Ohad Shaked, Liat Korn, Yair Shapiro, Avi Zigdon

**Affiliations:** 1School of Graduate Studies, Ariel University, Ariel 40700, Israel; 2Natali Healthcare Solutions, Ramat Gan 15208, Israel; 3Disaster Research Center IL, Ariel University, Ariel 40700, Israel; 4Department of Health Systems Management, School of Health Sciences, Ariel University, Ariel 40700, Israel; 5Health Promotion and Well-Being Research Center, Ariel University, Ariel 40700, Israel

**Keywords:** healthcare service, emergency healthcare service, older adult, COVID-19, social factors

## Abstract

This study examined social characteristics and their relations to healthcare service demand among older adults during the first COVID-19 lockdown in 2020. The sample was based on a cohort of 103,955 adults over the age of 65. A general index of needs was composed based on healthcare service use data and was predicted in a multi-nominal logistic regression. The frequency of the total needs significantly (*p* < 0.000) declined while supportive community services (4.9%, 2.0%), living in a community framework (27.0%, 15.2%), and living in a private residence (29.7%, 20.1%) were significantly associated (*p* < 0.000) with less frequent needs compared to the complementary groups. Supportive communities turned out to be an extremely important service for older adults. Policy makers should consider expanding supportive community services for older adults, as it was shown to have a positive correlation with lower healthcare service use, which might be an indicator of better overall health.

## 1. Introduction

COVID-19 severely impacted older adults, who had to be isolated in order to avoid exposure to the pandemic [1,2]. During the first lockdown, they were ordered not to go to work and were asked to recruit the help of family to purchase medicines and food. They were more strictly observed for responsiveness to isolation guidelines compared to the general population [1,3]. The social implications of the isolation instructions for this population were significant [4]. There is evidence of older adult people being at higher risk of social isolation and loneliness and, consequently, declining mental and physical health [5,6]. In addition, increased risk of negative physical health outcomes was observed [7], with some experiencing depression and anxiety in addition to an inability to obtain medication due to inadequate social support [8,9]. Older adult people and people with long-term health conditions tend to possess weak social networks and experience loneliness, resulting in more hospital and general practitioner visitations [10].

The impact of social factors among older adults as factors influencing the ability to prepare for catastrophic events has been investigated [11,12,13]. Various socioeconomic factors were associated with infection and mortality rates [12]. For example, social capital (which measures the extent of shared group resources within a community, the perception of fairness, perceived help, group membership, and trust) was negatively associated with mortality [14]. In particular, low socioeconomic status contributes to a higher incidence of diseases and, correspondingly, high socioeconomic status was found to contribute to lower mortality rates associated with social health, community support, and cohesion [12].

Loneliness and age are associated with the level of compliance with patient guidelines and the implementation of treatment recommendations [15]. Non-compliance with guidelines may lead to a deterioration in medical conditions while the implementation of medical guidelines significantly improves early detection of complications, with people living alone twice as likely to experience mortality from cardiovascular disease compared to those living with another person [15,16,17]. 

Social services and volunteering may play an important role in the mental health of older adults with no social support [9]. The higher the social involvement of the older adult in the community, the lower the risk for the individual. Social support is an important component in the well-being of the individual, with social cohesion describing how strong relationships are and whether there is a sense of solidarity among community members [12]. Social support may be very effective in the struggle against depression [18] and important to older adults as compensation for social disruption resulting from retirement. Establishing supportive community neighborhoods produces social support and aids with daily activities or the fulfilling of other basic needs such as responding to emergencies or accessing a physician. Supportive communities are of particular importance among older adults both in the context of sharing information about illnesses, informational support regarding risk factors and providing advice in case of complications, and providing information on available medical resources [19]. Additionally, the family is a resource that is increasingly needed by the older adult [20].

*Natali healthcare solutions* [21] is a private company that provides services to the older adult population in Israel, including medical services such as doctor home visits, ambulances, medical consultation telephone calls, and other various logistical services. If it is a life-threatening situation, an ambulance crew will be rushed to the patient’s address. 

Previous studies focused on social characteristics and their relations to healthcare service demand among older adults during the first COVID-19 lockdown. This study used a unique and unexplored dataset to look into this association, employing special social measures in order to draw a clearer picture of healthcare service provision during the time of the pandemic and thereby adding to the existing literature. The purpose of this study was to examine the contribution of different social factors to the healthcare service requirements during the first COVID-19 lockdown. In addition to socio-demographic characteristics, the specific social factors that were examined were living in a community framework, supportive community services, living in nursing homes, and type of housing. We hypothesized a decrease in the total medical needs during the first COVID-19 lockdown compared to the corresponding period in 2019, which would be more pronounced among those with different kinds of social support.

## 2. Methods

### 2.1. Procedure

This was a longitudinal observational study that followed a cohort of healthcare service provider customers. This study compared two periods: February-April 2019 (P1), before the beginning of the COVID-19 pandemic, and February-April 2020 (P2), during which the first COVID-19 lockdown took place in Israel. In P2, citizens of the State of Israel were subjected to COVID-19 lockdown restrictions, as promulgated by the Ministry of Health in Israel [22].

### 2.2. Population and Sample

*Natali healthcare solutions* has regular subscribers and provides subscribers with medical services according to the type of service to which the subscriber is registered. The healthcare service provider provides a panic (emergency) button service for the call center, doctor home visit services, an ambulance service, and remote telephone consultation. All customers that constituted the population of this study were adults over the age of 65 in Israel. The sample did not undergo a sampling process but was based on all relevant healthcare service provider data. No identification data of the subjects was presented in the research data file. For this study, customers under the age of 65 were removed from the dataset to include only older adults during the two periods under study [23]. A lack of data on customers in one of the two periods examined led to the removal of some customers from the sample.

Table 1 shows that a total of 103,955 Natali customers were included in the sample, representing the entire relevant aged customer population. The sample included 63.5% women (n = 65,954) and 35.0% men (n = 36,364). For 1.6%, there was no information regarding gender (n = 1637). Most of the sample were married (61.6%), Jewish (83.9%), and secular (98.4%).

### 2.3. Measures

*Natali healthcare solutions* collects data for the purpose of monitoring and supervising healthcare service provider activities. The healthcare service provider data was provided solely for the purpose of the present study, excluding identifying information of the members. Most of the variables relevant to the present study were derived from the healthcare service provider data while other residential variables were based on the CBS. Socio-economic status, level of religiosity, and residential sector were the only relevant sociodemographic characteristics that were provided from the CBS [24] after matching with the member’s residential address.

In situations where there is no risk to human life, the center provides telephone counseling or offers a doctor for a home visit. A supportive community service is also offered in which customers are contacted on a weekly basis for assistance with their needs: questions, consultation, purchase of medicines, purchase of food, and any logistical or medical needs. 

The aim of the present study was to examine the social characteristics and their impact on the demand for medical services among older adults in Israel during the COVID-19 lockdown in 2020 compared to the corresponding period in 2019. We hypothesized that there would be a decrease in the demand for medical needs during the first COVID-19 lockdown period, which would be more pronounced among those with different kinds of social support.

### 2.4. Description of the Variables

#### 2.4.1. Dependent Variable

The needs for services index constituted the dependent variable for each period separately: pre-COVID-19 (P1) and during the first COVID-19 lockdown (P2). A general index of needs was composed based on four variables: *1. Medical* calls—The number of times a customer contacted the emergency call center for a medical reason. The value scale was P1 0-114 and P2 0-472. *2. Emergency* calls—The number of times a customer contacted the emergency call center. The value scale was P1 0-226 and P2 0-294. *3. Ambulance order*—The number of times the customer ordered an ambulance in both periods. These created a scale of values as detailed: P1 0-18 and P2 0-16. *4. Doctor home visit*—The number of times a client called a doctor during these periods. The scale of values was P1 0-102 and P2 0-111. These four variables once indexed had an internal consistency of Cronbach alpha α = 0.693 at P1 and α = 0.630 at P2. To construct this index, the four variables were first recoded as dichotomous variables that received values of 0—no service required and 1—required service, and then the four were summed into one complex index for each period on a scale of 0–4. As these four variables had different scales, the action to dichotomize (creating 2 groups of 0—no need or 1—one need or more) had to be implemented in order to unite different scales into one before combining them into one index. 

#### 2.4.2. Independent Variables

Our social independent variables were tested. *1. Supportive community* services—A service of medical and logistic assistance on a weekly basis. Values are yes—service is supplied or no—not supplied. *2. Living in nursing* homes—Whether the service consumer lives in an older adult home (yes/no). *3. Living in community* framework—This was taken from the service consumer’s home address and matched with the Israeli CBS [24] information (yes/no). *4. Type of housing*—(Private residence/building or apartment) If the floor number of the home address was supplied, the type of housing was considered a building. All others were considered private homes. 

#### 2.4.3. Socio-Demographic Variables

Sex—Male/female. *Age*—This variable was calculated at the starting point of the study (2019) by year of birth. The ages of two participants were missing. In a new age scale, the age group was grouped into three *age categories* [23,25]: young adults (65–75), middle-aged adults (76–85), and older adults (86 and older). For regression analysis, the age scale was divided by the median to 0 = younger and 1 = older. *Marital status*—Single/married/divorced/widowed, although 28,742 service users (27.5% of the sample) had a missing marital status. For regression analysis, family status was married and the three values of not married (single, divorced, widower) were combined. Three additional socio-demographic variables were derived from the respondent’s residential address after cross-referencing the information with CBS data [24]: *Residential sector*—The ethnicity of the community by service user address (1. Jewish, 2. Non-Jewish, and 3. Mixed); *Residential religiosity* of the locality by service user address (1. Secular locality, 2. Religious locality, and 3. Ultra-Orthodox locality); and the *Socio-economic status* of the locality according to CBS coding [24] in which communities were classified into clusters from 1–10, with 1 symbolizing a very low socioeconomic status and 10 a very high socioeconomic status. This scale was divided into three categories: 1. Low class—clusters 1–5; 2. Middle class—clusters 6–7; and 3. High class—clusters 8–10.

### 2.5. Data Analysis

Data was analyzed using IBM SPSS Statistics 25. First, descriptive statistics for describing the sample by frequency were presented. Then, cross tabulation frequencies between needs in P1 and P2 based on the independent social variables, with a Chi square test for significant differences between groups were implemented. At last, multi-nominal logistic regression results for predicting needs for services in two models: 1. For the comparison of no needs as a reference group for one need and 2. For the comparison of no needs as a reference group for two or more needs. For this purpose, the dependent variable “need for services” was dichotomized by the median. The expected β values shown in the table constitute the RR or relative risk and show the probability of service needs according to the groups of independent variables. The confidence interval (CI) displays the minimum value and the maximum value for the 95% confidence range. Values displayed in bold were significantly higher or lower than 1.

## 3. Results

A large older adult sample that contained 103,955 members of *Natali healthcare solutions* and their needs for medical services during pre-pandemic (P1) and COVID-19 lockdown period (P2) were checked according to social variables. 

Table 2 presents the distribution of needs for services in the two studied periods. All differences between groups were found to be significant (*p* < 0.001), and a decline in needs for services was observed from P1 to P2. From all kinds of housing, living in nursing homes required more services in both study periods (47.8% and 37.4%, respectively) and supportive community services required less services in both study periods (4.9% and 2.0%, respectively). Supportive community services were found to be significantly different between groups of needs in both studied periods. The frequency of needs was lower for customers who used supportive community services in P1 (4.9%) compared to those who did not use them (33.4%) and went down even more in P2 (2.0%) during the COVID-19 lockdown for customers who used supportive community services compared to those who did not (24.2%). For those who live in nursing homes, the frequency of needs was 47.8% in P1 compared to those who do not live in nursing homes (27.4%). Needs for services declined in P2 to 37.4% during the COVID-19 lockdown for customers who live in nursing homes compared to those who do not live in nursing homes (18.9%). In sum, Table 2 shows that in P1 and P2, supportive community services, living in a community framework, and living in a private residence were significantly associated with less frequent needs, although living in nursing homes was associated with a higher frequency of needs.

Table 3 presents a multi-nominal logistic regression to predict needs for services during COVID-19 lockdown periods in two models of comparison: 1. Zero needs to one need and 2. Zero needs to two or more needs. In this analysis, 73,127 participants had all their data examined and the explained variance was 33.3%. All nine variables were significant in model 1 while in model 2, only gender was insignificant. In model 1, need in P1 was the strongest predicting variable (RR = 8.555, 95% CI 8.178–8.950), followed by supportive community services (RR = 3.946, 95% CI 2.475–6.290) and nursing homes (RR = 2.002, 95% CI 1.912–2.097). The analysis shows that the probability of having at least one need during the COVID-19 lockdown was higher for males rather than females (RR = 0.937, 95% CI 0.893–0.984), for older service users than younger (RR = 1.173, 95% CI 1.115–1.235), for not married than married (RR = 1.354, 95% CI 1.290–1.421), for those who do not live in community frameworks (RR = 1.604, 95% CI 1.361–1.891), and for those who live in buildings rather than private residences (RR = 1.107, 95% CI 1.041–1.178).

Model 2 shows results in the same direction. The strongest predicting variables were previous needs, supportive community services, and living in a community framework. The probability of having 2 or more needs was 24 times higher if there were needs the year before (RR = 24.548, 95% CI 22.733–26.724). The probability of two needs or more was significantly higher (RR = 11.390, 95% CI 3.617–35.871) for those who did not have supportive community services. The probability of two needs or more was significantly higher (RR = 1.999, 95% CI 1.555–2.568) for those who did not live in a community framework.

## 4. Discussion

Epidemics cause social and medical problems and make it imperative to respond quickly during times of emergency [12]. This study characterized the consumption of services among older adults in Israel during the first COVID-19 lockdown in 2020 compared to the corresponding period of the previous year in 2019 according to various social characteristics: Supportive community services, living in nursing homes, living in a community framework, and type of housing, based on a large older adult sample that contained 103,955 members of *Natali healthcare solutions.* The sample included 63.5% women and the average age of the participants was 80 years old (standard deviation = 7.46). Regarding family status, 61.6% were married and 30.9% were widowers. Most of the sample were living in Jewish (83.9%) and secular (98.4%) communities. According to the Israeli Central Bureau of Statistics [23], participants were clustered by residential socioeconomic status to 37.6% below average, 37.2% average, and 25.2% above average, with only 2.9% (n = 3001) living in what is considered a community framework by the Israeli CBS (2019). Only 2.2% of the sample (n = 2295) purchased the supportive community services supplied by *Natali healthcare solutions* while around one-quarter of the sample (26.2%, n = 27,277) were living in nursing homes. Regarding the type of housing, 20% (n = 20,266) were living in private homes. 

We hypothesized a decrease in the total medical needs during the first COVID-19 lockdown compared to the corresponding period in 2019, which would be more pronounced among those with different kinds of social support.

An examination of the findings according to the two study periods showed that during the period of the first COVID-19 lockdown, the frequency of older adult people applying to *Natali healthcare solutions* for various social and medical matters was significantly lower compared to the corresponding period in 2019, as was expected. The frequency of the total needs significantly (*p* < 0.000) declined in P2 (23.8%) compared to P1 (32.8%). In times of crisis, older adults tend to avoid being examined and going to hospitals, preferring to deal with their medical condition on their own [26,27,28].

One of the services supplied by *Natali healthcare solutions* is a supportive community, in which service users are contacted on a weekly basis to assist them with questions, consultation, purchase of medicines, purchase of food, and any logistical or medical needs they may have. Our findings show that those who made use of this supportive community service were significantly less likely to demand medical services in both periods. In addition, living in a community framework was significantly associated with less frequent needs. These findings are supported by previous studies [9,12] demonstrating higher social involvement being associated with higher well-being. Community support highlights the particular importance for older adults of information sharing about diseases, informational support for risk factors, advice in dealing with complications, and providing information on available medical resources [19]. Correspondingly, isolation from social environment was associated with less effective disease management and deterioration [29,30,31].

The older adult population living in nursing homes represents a high-risk group [32]. This may explain the finding in the current study that nursing home residency doubled the probability of needs compared to living in private residences. Older adults who live on their own avoided going to hospitals, attempting to fend for themselves [26,27,28] as opposed to those living in nursing homes. Support for this can be found in studies dealing with infections in nursing homes. Nursing homes had high rates of infectious disease due to overcrowding, shared public facilities, shared services, and low levels of infection control. They also had to deal with the consequences of isolation, contact restrictions, and social distance. Although preventive and control measures should be taken to prevent viruses from entering a facility, there was greater viral spread in the nursing homes than in the community [33,34,35]. Moreover, it may be that nursing home staff encouraged medical services due to the increase in demand for medical needs [36], especially during the first COVID-19 closure. This may explain the increased consumption of services in nursing homes.

The type of housing also has an impact on the consumption of healthcare services. This study found that supportive community services and living in a community framework are protective factors. Regarding differences between types of housing, a few studies found that isolation was particularly difficult in crowded apartments inhabited by low-income individuals, including older adults and those most susceptible to infection. As a result, limited living space contributed to the exacerbation of disease and mental disorders, mainly due to overcrowding [37,38,39,40].

## 5. Conclusions

Social variables were found to be significantly associated with lower consumption of healthcare services during the pandemic. Supportive communities turned out to be an extremely important service for older adults, and we recommend that this service is expanded. Policymakers should consider expanding supportive community services for older adults, as it was shown to have a positive correlation with lower healthcare service use, which may be an indicator of better overall health. Further studies should look into the differences between nursing homes and older adult homes that are non-nursing homes, which may be able to explain these findings.

This study was based on a very large cohort of older adult people in Israel and designed prospectively to check for risk and protective factors associated with medical needs. Its design and sample allow for stronger confidence in the conclusions of this paper. The limitations of this study prevent us from positing causal relations between the variables as it is observational in nature and cannot prevent the interference of confounder variables. Another limitation may be due to the medical condition of those who live in nursing homes compared to older adult people living in their homes. It is possible that those who live in a nursing home in the first place have worse medical conditions than those living in their homes. A final limitation concerns the uncertainty in the variable of nursing homes since data on differences between nursing homes and older adult homes that are non-nursing homes were not available. 

This paper presents the need for healthcare services of older adults in Israel during the first COVID-19 lockdown. It reinforces the claim that in times of crisis, older adults tend to avoid receiving healthcare services and prefer to deal with their medical conditions on their own. The main finding is that older adults in supportive community services were significantly less likely to require medical services during the first lockdown. Supportive community services should be expanded to lower the need of older adults for healthcare services, in routine times and in times of crisis. 

## Figures and Tables

**Table 1 healthcare-10-01854-t001:** Sample description by socio-demographic variables.

Variables	Values	Total Sample	
		N	%
		103,955	100
Gender	Women Men Unknown	65,954 36,364 1637	63.5 35.0 1.6
Age categories	65–75 76–85 86 and older	30,252 47,846 25,855	29.1 46.0 24.9
Family status	Single Married Divorce Widower	1105 46,360 4496 23,252	1.5 61.6 6.0 30.9
Residential sector	Jewish Not Jewish Mixed	87,210 918 15,825	83.9 0.90 15.2
Residential religiosity	Secular Religious Ultra-Orthodox	102,305 421 1229	98.4 0.40 1.2
Residential socioeconomic status (SES)	Below average (clusters 1–5) Average (clusters 6–7) Above average (clusters 8–10)	39,124 38,677 26,152	37.6 37.2 25.2
Living in community framework	Yes No	3001 100,954	2.9 97.1
Supportive community services	Yes No	2295 101,660	2.2 97.8
Living in nursing homes	Yes No	27,277 76,678	26.2 73.8
Type of housing	Private Building/apartment	20,266 81,168	20.0 80.0

**Table 2 healthcare-10-01854-t002:** Need for services pre-pandemic (P1) and in the COVID-19 lockdown period (P2) by social variables.

Social Variable	Values	P1-Pre-Pandemic Need for Services		n	* Sig. X^2^	P2-COVID-19 Lockdown Need for Services		n	* Sig.
		No Needs	At Least Once			No Needs	At Least Once		
**Supportive community services**	**Yes**	95.1%	4.9%	2295	0.000 829.7	98.0%	2.0%	2295	0.000 610.6
	**No**	66.6%	33.4%	101,660		75.8%	24.2%	101,660	
**Living in nursing homes**	**Yes**	52.2%	47.8%	27,277	0.000 3791.2	62.6%	37.4%	27,277	0.000 3800.6
	**No**	72.6%	27.4%	76,678		81.1%	18.9%	76,678	
**Living in a community framework**	**Yes**	73.0%	27.0%	3001	0.000 47.2	84.8%	15.2%	3001	0.000 126.0
	**No**	67.0%	33.0%	100,954		76.0%	24.0%	100,954	
**Type of housing**	**Private**	70.3%	29.7%	20,266	0.000 109.3	79.9%	20.1%	20,266	0.000 179.3
	**Building**	66.4%	33.6%	81,168		75.4%	24.6%	81,168	
**Total needs**		**67.2%**	**32.8%**	**103,955**	**0.000** **829.7**	**76.2%**	**23.8%**	**103,955**	**0.000** **26,159.3**

* Significance of Pearson chi-square for differences between independent groups of social variables, *p* < 0.001.

**Table 3 healthcare-10-01854-t003:** Multi-nominal logistic regression to predict need for services during COVID-19 lockdown periods.

Variables	Values	P2-COVID-19 Lockdown Need for Services					
		Model 1—One Need (Ref 0)			Model 2-Two or More Needs (Ref 0)		
		Exp β	CI		Exp β	CI	
			Lower	Upper		Lower	Upper
Gender	0 = male, 1 = female	**0.937 ****	0.893	0.984	1.013	0.947	1.083
Age	0 = younger, 1 = older	**1.173 *****	1.115	1.235	**1.513 *****	1.414	1.619
Family status	0 = married, 1 = not married	**1.354 *****	1.290	1.421	**1.256 *****	1.175	1.342
Residential socioeconomic status (SES)	0 = average, 1 = lower or upper	**1.068 ****	1.022	1.117	**1.080 ***	1.016	1.148
Needs in P1	0 = less, 1 = more	**8.555 *****	8.178	8.950	**24.648 *****	22.733	26.724
Supportive Community Services	0 = yes, 1 = no	**3.946 *****	2.475	6.290	**11.390 *****	3.617	35.871
Living in nursing homes	0 = no, 1 = yes	**2.002 *****	1.912	2.097	**1.256 *****	1.178	1.340
Living in community framework	0 = yes, 1 = no	**1.604 *****	1.361	1.891	**1.999 *****	1.555	2.568
Type of housing	0 = private home, 1 = Building/apartment	**1.107 ****	1.041	1.178	**1.105 ***	1.016	1.202
Nagelkerke R-Square		33.3%					
n		73,127					

** p <* 0.05, *** p <* 0.01, **** p <* 0.001; SES—Socio-Economic Status; SCS—Supportive Community Services. Significant values were bolded.

## Data Availability

Data belong to Natali Health Care Solutions and cannot be shared.

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
