# Peer review of "Social Factors Contributing to Healthcare Service Requirements during the First COVID-19 Lockdown among Older Adults"

_healthcare, 2022, doi:10.3390/healthcare10101854_

Round 1
Reviewer 1 Report
line 67 "privet compony provide services to the older adult" to correct
check the English in all the manuscript
the description of table 1 is too long, I think it could be better the describe the table in the discussion.
tab 2 change tab because it's not so clear
the description of the results and discussion could be more understandable
Author Response
Reviewer #1 comments
Reviewer 1: line 67 "privet compony provide services to the older adult" to correct
Author’s response: Done as suggested.
Reviewer 1: check the English in all the manuscript
Author’s response: The manuscript had been checked for English as requested, and a few changes were implemented.
Reviewer 1: the description of table 1 is too long, I think it could be better the describe the table in the discussion.
Author’s response: The description of table 1 had been shortened in the method section and was added in the discussion, as suggested.
Reviewer 1: tab 2 change tab because it's not so clear
Author’s response: Agreed. We provided a different design.
Reviewer 1: the description of the results and discussion could be more understandable
Author’s response: Following the reviewer comment, we went over these sections and improved the wording in a few places.

Reviewer 2 Report
The manuscript presents several grammatical issues that make it difficult to understand. In addition, there is not a clear definition of the research gap to be covered. Also, the methodology does not clarify some steps such as the scale definition used for the dependent variable. Moreover, several sentences are repeated in the text. Therefore, these issues make the manuscript unsuitable for publication in its actual form.
The following are some issues examples:
“We followed a cohort of 103,955 adults over the age of 65, and a general index of needs was composed based on healthcare service use data and was predicted in a multi-nominal logistic regression.”
In the series of elements joined together by coordinating conjunctions, the individual elements of the series are not equal (two elements have a subject one does not; one element is in the active voice, two in the passive voice)
“The higher the social involvement of the older adult in the community, the lower the risk for the individual and social support is an important component in the well-being of the individual”
The comparative structure that begins this sentence is confusing. The way it is joined to the main message of the sentence does not flow smoothly. This type of comparative structure probably would best be expressed in a more formal style.
Authors stated “For this study, customers under the age of 65 were removed from the dataset to include only the older adults during the two periods under study,”
What is the rationale for this decision and this age?
Authors stated “Table 1 shows that a total of 103,955 Natali customers participated in the sample, representing the entire relevant aged customer population”
This might be confusing. Customers did not participate in the study. As far as I understand, customers were included by the authors. Therefore, the authors must clarify this redaction and, in case of sensitive data from the participants being included, an ethical committee approval should be submitted along with this manuscript.
The note for Table 1 is long. Most of the information is self-described in Table 1. A few clarifications might be included within the text instead of including it as a note.
Text in rows 121-124 is repetitive from the previous text. Please avoid this in order to increase clarity.
Table 2 looks bulky; some words or numbers are cut or moved.
Authors used different font styles in the text, the following is an example “A large older adult sample that contained 103,955 members of Natali healthcare solutions, needs for medical services during pre-pandemic (P1) and Covid-19 lockdown period (P2) were checked according to social variables.”
The following sentences are confusing
“All customers constitute the population of this study- adults over the age of 65 in Israel.”
“Natali healthcare solutions is a privet compony provide services to the older adult population in Israel,”
Author Response
Reviewer #2 comments
Reviewer 2: The manuscript presents several grammatical issues that make it difficult to understand. In addition, there is not a clear definition of the research gap to be covered. Also, the methodology does not clarify some steps such as the scale definition used for the dependent variable. Moreover, several sentences are repeated in the text. Therefore, these issues make the manuscript unsuitable for publication in its actual form.
Author’s response: We would like to thank the reviewer for his/her comments and examples. We did our best to improve the manuscript.
Reviewer 2: The following are some issues examples:
“We followed a cohort of 103,955 adults over the age of 65, and a general index of needs was composed based on healthcare service use data and was predicted in a multi-nominal logistic regression.”
In the series of elements joined together by coordinating conjunctions, the individual elements of the series are not equal (two elements have a subject one does not; one element is in the active voice, two in the passive voice)
Author’s response: Thank you. We reworded the sentence.
Reviewer 2: “The higher the social involvement of the older adult in the community, the lower the risk for the individual and social support is an important component in the well-being of the individual”
The comparative structure that begins this sentence is confusing. The way it is joined to the main message of the sentence does not flow smoothly. This type of comparative structure probably would best be expressed in a more formal style.
Author’s response: Agreed. Reworded as suggested.
Reviewer 2: Authors stated “For this study, customers under the age of 65 were removed from the dataset to include only the older adults during the two periods under study,”
What is the rationale for this decision and this age?
Author’s response: Justification for this decision was added to the text as follows (please see reference 23): Im Ryu S, Cho B, Chang SJ, Ko H, Yi YM, Noh EY, Cho HR, Park YH. Factors Related to Self-Confidence to Live Alone in Community-Dwelling Older Adults: A Cross-Sectional Study. BMC geriatrics. 2021 Dec;21(1):1-2.).
Reviewer 2: Authors stated “Table 1 shows that a total of 103,955 Natali customers participated in the sample, representing the entire relevant aged customer population”
This might be confusing. Customers did not participate in the study. As far as I understand, customers were included by the authors. Therefore, the authors must clarify this redaction and, in case of sensitive data from the participants being included, an ethical committee approval should be submitted along with this manuscript.
Author’s response: All customers of the company were included in the data file by the authors unless they did not match the inclusive criteria. No identification data of the subjects was presented in the research data file. The sentence was reworded with an explanation added in the text (line 88).
Reviewer 2: The note for Table 1 is long. Most of the information is self-described in Table 1. A few clarifications might be included within the text instead of including it as a note.
Author’s response: Thank you. Redundant text was removed as suggested.
Reviewer 2: Text in rows 121-124 is repetitive from the previous text. Please avoid this in order to increase clarity.
Author’s response: Thank you. Redundant text was removed as suggested.
Reviewer 2: Table 2 looks bulky; some words or numbers are cut or moved.
Author’s response: Table 2 was rearranged to present the numbers more clearly.
Reviewer 2: Authors used different font styles in the text, the following is an example “A large older adult sample that contained 103,955 members of Natali healthcare solutions, needs for medical services during pre-pandemic (P1) and Covid-19 lockdown period (P2) were checked according to social variables.”
Author’s response: Noticed and corrected. We carefully checked the entire manuscript for font change.
Reviewer 2: The following sentences are confusing
“All customers constitute the population of this study- adults over the age of 65 in Israel.”
Author’s response: Text reworded to clarify the original meaning of the sentence.
Reviewer 2: “Natali healthcare solutions is a privet compony provide services to the older adult population in Israel,”
Author’s response: Text reworded to clarify the original meaning of the sentence.
Reviewer 3 Report
This study is to explore socioeconomic factors affecting healthcare service requirements during the first COVID-19. The topic is interesting and timely. Overall study processes are proper and valid. Following are some comments to improve the quality of the manuscript.
- Address some contributions of this study in the introduction. For example, There are many previous studies related to this concern, however few studies are explored ....
- Address the study of purpose in the introduction.
- In line 67, compony => company.
- In line 67, if possible, do not mention the company name since you mention it in line 83.
- Lines 94 - 107, no need to make bold type. Make them plain.
- company => I would like to ask you to change the word "company" to a healthcare service provider.
- In 2.2. Population and sample, mention a data validity. Who participates to validate data collected and how they validate the data. One or two sentences are enough.
- Provide each chi-square in each p-value accordingly.
- It is not necessary to make an independent section of 6. Strengths and limitations. Delete the section title so that it becomes a part of the conclusion.
- In lines 311 - 320, if they are part of the manuscript, delete the subtitles and make sentences.
Author Response
comments
Reviewer 3: This study is to explore socioeconomic factors affecting healthcare service requirements during the first COVID-19. The topic is interesting and timely. Overall study processes are proper and valid. Following are some comments to improve the quality of the manuscript.
Author’s response: Thank you.
Reviewer 3: Address some contributions of this study in the introduction. For example, There are many previous studies related to this concern, however few studies are explored ....
Author’s response: Study contribution was added at the end of the Introduction as suggested (lines 71-73) as follows: “Previous studies focused on social characteristics and their relations to healthcare service demand among older adults during the first COVID-19 lockdown. This study used a unique and unexplored dataset to check look into this association, employing special social measures in order to draw a clearer picture of healthcare service provision during a time of pandemic and thereby adding to the existing literature.”
Reviewer 3: Address the study of purpose in the introduction.
Author’s response: Added as requested.: “The purpose of this study was to examine the contribution of different social factors to the healthcare service requirements during the first COVID-19 lockdown. Beside socio-demographic characteristics, the specific social factors that were examined were living in community framework, supportive community services, living in nursing homes and type of housing.”
Reviewer 3: In line 67, compony => company.
Author’s response: Thank you. Spelling was corrected.
Reviewer 3: In line 67, if possible, do not mention the company name since you mention it in line 83.
Author’s response: Removed from line 83.
Reviewer 3: Lines 94 - 107, no need to make bold type. Make them plain.
Author’s response: Altered as suggested.
Reviewer 3: company => I would like to ask you to change the word "company" to a healthcare service provider.
Author’s response: Altered as suggested
Reviewer 3: In 2.2. Population and sample, mention a data validity. Who participates to validate data collected and how they validate the data. One or two sentences are enough.
Author’s response: The database was collected for the work needs of the Natalie Medical Services provider: variables were checked and validated against the Central Bureau of Statistics, and the research variables were collected and validated from a sample against the existing reports in the history and medical files of the subscribers.
Reviewer 3: Provide each chi-square in each p-value accordingly.
Author’s response: Added as requested.
Reviewer 3: It is not necessary to make an independent section of 6. Strengths and limitations. Delete the section title so that it becomes a part of the conclusion.
Author’s response: Title “Strengths and limitations” was deleted as requested.
Reviewer 3: In lines 311 - 320, if they are part of the manuscript, delete the subtitles and make sentences.
Author’s response: Done as requested.
Round 2
Reviewer 1 Report
Ok
Reviewer 2 Report
I would like to congratulate the authors for their efforts. The manuscript in its new version contains all the suggestions and corrections identified. I consider that the article can be accepted for publication.